# An inherently interpretable AI model improves screening speed and accuracy for early diabetic retinopathy

Kerol Djoumessi[1,2], Ziwei Huang[1,2], Laura Kühlewein[3], Annekatrin Rickmann[3,4], Natalia Simon[5], Lisa M. Koch[1,2,6], Philipp Berens[1,2*‡]

**1** Hertie Institute for AI in Brain Health, University of Tübingen, Tübingen, Germany, **2** Tübingen AI Center, University of Tübingen, Tübingen, Germany, **3** University Eye Hospital, University of Tübingen, Tübingen, Germany, **4** Eye Clinic Sulzbach, Knappschaft Hospital Saar, Sulzbach, Germany, **5** Black Forest Eye Clinic, Endingen, Germany, **6** Department of Diabetes, Endocrinology, Nutritional Medicine and Metabolism UDEM, Inselspital, Bern University Hospital, University of Bern, Bern, Switzerland

‡ Current address: Hertie Institute for AI in Brain Health, University of Tübingen, Tübingen, Germany
* philipp.berens@uni-tuebingen.de

**Data availability statement:** The implementation of our sparse BagNet model is

## Abstract

Diabetic retinopathy (DR) is a frequent complication of diabetes, affecting millions worldwide. Screening for this disease based on fundus images has been one of the first successful use cases for modern artificial intelligence in medicine. However, current state-of-the-art systems typically use black-box models to make referral decisions, requiring post-hoc methods for AI-human interaction and clinical decision support. We developed and evaluated an inherently interpretable deep learning model, which explicitly models the local evidence of DR as part of its network architecture, for clinical decision support in early DR screening. We trained the network on 34,350 high-quality fundus images from a publicly available dataset and validated its performance on a large range of ten external datasets. The inherently interpretable model was compared to post-hoc explainability techniques applied to a standard DNN architecture. For comparison, we obtained detailed lesion annotations from ophthalmologists on 65 images to study if the class evidence maps highlight clinically relevant information. We tested the clinical usefulness of our model in a retrospective reader study, where we compared screening for DR without AI support to screening with AI support with and without AI explanations. The inherently interpretable deep learning model obtained an accuracy of .906 [.900–.913] (95%-confidence interval) and an AUC of .904 [.894–.913] on the internal test set and similar performance on external datasets, comparable to the standard DNN. High evidence regions directly extracted from the model contained clinically relevant lesions such as microaneurysms or hemorrhages with a high precision of .960 [.941–.976], surpassing post-hoc techniques applied to a standard DNN. Decision support by the model highlighting high-evidence regions in the image improved screening accuracy for difficult decisions and improved screening speed. This shows that inherently interpretable deep learning models can provide clinical decision support while obtaining state-of-the-art performance improving human-AI collaboration.

---

available at GitHub (https://github.com/
kdjoumessi/Sparse-BagNet_clinical-validation).
The annotations performed for this study on
selected Kaggle database images, the study
data, and the analysis are available in the same
GitHub repository.

**Funding:** This work was supported by a grant of
the Hertie Foundation to PB, grants from the
German Research Foundation (BE5601/8-1 to
PB; Excellence Cluster 2064 "Machine Learning
— New Perspectives for Science", project
number 390727645 to PB), a grant from the
Carl Zeiss Foundation ("Certification and
Foundations of Safe Machine Learning Systems
in Healthcare" to LK). Furthermore, the
International Max Planck Research School for
Intelligent Systems (IMPRS-IS) supported KD.
The funders had no role in study design, data
collection and analysis, decision to publish, or
preparation of the manuscript.

**Competing interests:** The authors have
declared that no competing interests exist.

## Author summary

AI systems designed to support clinical decision making use black-box deep learning
models for many medical applications. This includes AI-based screening systems for
diabetic retinopathy, a sight threatening complication of diabetes. This hinders clinical
uptake of such methods, as clinicians and patients do not have a way to validate the AI
systems decisions. Sometimes, post-hoc methods are used to generate heatmaps that
supposedly explain the AI systems decision. However, these methods are problematic
as the generated explanations do not reflect the actual decision-making process of the
model and are prone to spurious correlations. In our paper, we take a big step forward
for enabling trustworthy AI systems for supporting clinical decision making in screen-
ing for diabetic retinopathy: We introduce an inherently interpretable deep learning
model which provides human-understandable explanations for its decisions. The model
combines the power of deep learning with the interpretability of simpler models such
as logistic regression by computing an explicit evidence map. This map forms the basis
of the model's decisions, alleviating the issues of post-hoc techniques. We validate the
clinical potential of this model for improving diabetic retinopathy screening showing
highlighting regions with high disease evidence during clinical grading decreased the
grading time significantly and improved grading accuracy for difficult borderline cases.

## Introduction

Diabetic retinopathy (DR) screening has been one of the first successful use cases for artificial
intelligence (AI) in medicine [1], promising fast, cost-effective access even where insufficient
clinical personnel is available. By now, multiple AI systems have received regulatory clearance
[2,3] and have been found useful to triage patients not requiring specialist attention and those
with vision-threatening DR, potentially contributing to increased screening adherence [4].

However, current state-of-the-art models use black-box deep learning approaches to make
referral decisions, providing clinicians only with limited binary recommendations to either
refer a patient for further examination or not. Yet, the performance of current systems still
typically makes some level of human grader verification necessary [3], which could be guided
by an useful explanation of the AI system's decision. Also, clinical implementation would
benefit from clinicians being able to understand the rationale behind the recommendation of
the algorithm [5–7].

Typically, an AI system's decision are explained with heatmaps obtained post-hoc using
gradient-based approaches [8–10]. However, such explanations are not trustworthy, as the
produced heatmaps do not reflect the actual decision-making process of the model, and are
prone to spurious correlations [11]. Therefore, their results cannot be easily integrated into
the clinical decision-making process [7,12].

We address this issue and validate an inherently interpretable deep learning architecture
for providing clinical decision support for screening for early DR in a retrospective reader
study. Our approach uses a deep learning architecture called sparse BagNets [13,14], which
explicitly models the local evidence for the presence of DR as part of its network architec-
ture (Fig 1B). Most studies so far have considered the task of screening for moderate non-
proliferative DR or more advanced stages [1], although even mild non-proliferative dia-
betic retinopathy (NPDR) is recommended for close monitoring and careful control of
hyperglycemia [15,16]. We reasoned that the benefit of AI-based explanations and decision
support would be most clearly visible for this challenging diagnostic task. Trained on a large

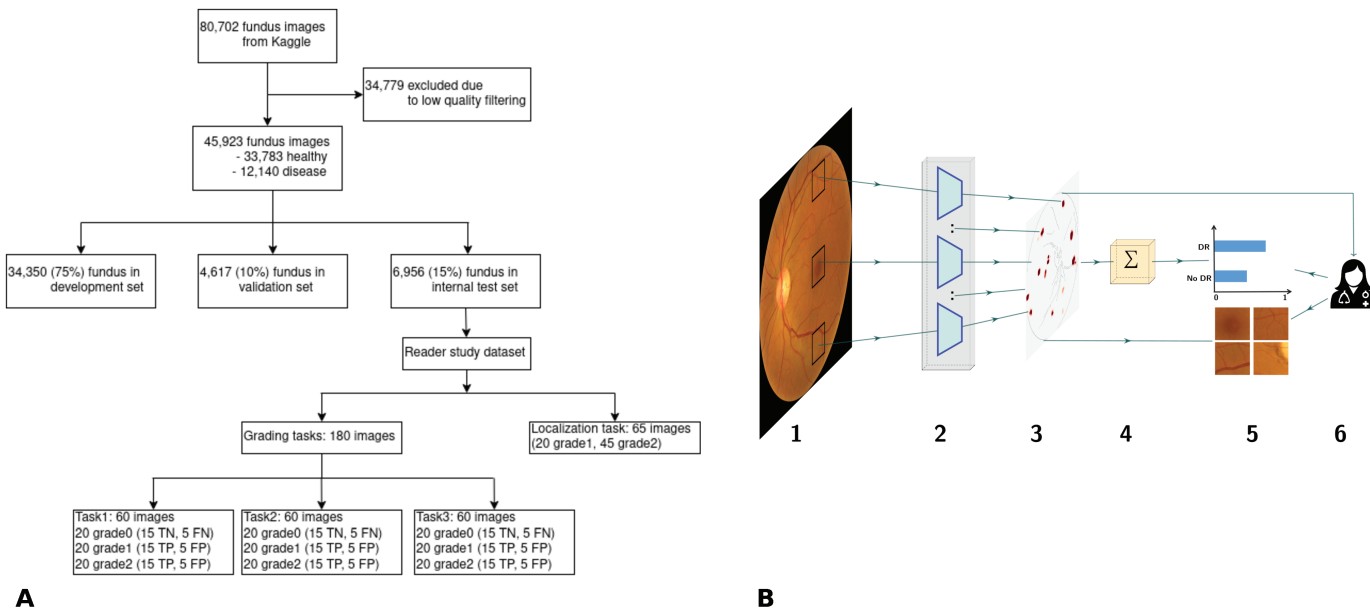

**Fig 1. Overview of the development data and proposed inherently interpretable deep learning framework evaluated in this study.** (A) Summary of the development dataset used to build the model, as well as the data used in the retrospective reader study. (B) Sparse BagNet architecture. 1. As a preliminary step, the retinal fundus image is implicitly split into many overlapping small patches of size 33 × 33. 2. All patches are fed to the model backbone, which processes them in parallel. 3. The BagNet backbone generates a heatmap that depicts the local disease evidence of individual patches. 4. The values of the heatmap are averaged and used as the final logit for classification. 5./6. The logits are fed into a softmax function which provides the probability distribution of the output, and then patches of suspect regions based on the heatmaps can be requested and viewed by a clinician to understand the classification results.

publicly available dataset, our model shows high specificity and sufficient sensitivity in detecting mild DR across a large array of datasets. Importantly, we show that the obtained class evidence maps highlight clinically relevant lesions such as microaneurysms or hemorrhages with high precision, making them useful for verifying the AI system's decisions. Finally, we show that the system can be effectively used to guide clinical decision-making, leading to 17.5% improvement in diagnostic accuracy for mild DR and overall about ≈ 25% improvement in screening time.

## Methods

### Dataset description and data preparation

We used eleven publicly available retinal image datasets, consisting of color fundus images from various sources, to develop and evaluate an inherently interpretable deep learning model for early DR detection (Table 1). For all datasets, fundus images had assigned reference grades based on the International Clinical Diabetic Retinopathy classification scale [17], which provides a grading scheme ranging from 0 (no DR), 1 (mild NPDR), 2 (moderate NPDR), 3 (severe NPDR) to 4 (proliferative DR) according to DR severity. As our goal was to develop an AI system for early DR screening, we combined class level {0} vs {1,2,3,4}. At stage 1, DR is in most cases asymptomatic, and challenging to detect even for experienced ophthalmologists. As all fundus datasets were fully anonymous, no approval from an Ethics Board was needed for this part of the study.

**Table 1. Summary of the internal and external validation datasets used to evaluate the models. "Origin" refers to the country where the data was collected. "Lesion" refers to the number of images in the dataset with lesion annotations. The Kaggle dataset (first row, shaded in gray) is the internal dataset used to evaluate the model, while the other datasets were used for external validation to assess the generalization properties of the trained model.**

| Dataset | Origin | Number of images | | | Lesion |
|---|---|---|---|---|---|
| | | All | Healthy | DR | |
| Kaggle [18] | USA | 6,956 | 5,118 | 1,838 | 65 |
| IDRiD [19] | India | 512 | 168 | 348 | 81 |
| E-Ophtha [20] | France | 434 | 260 | 174 | 174 |
| FGA-DR [21] | UAE | 1,841 | 101 | 1,740 | 1,740 |
| DIARETDB1 [22] | Finland | 89 | 05 | 84 | 84 |
| DDR [23] | China | 12,513 | 6,265 | 6,248 | 755 |
| DR2 [24] | Brazil | 445 | 300 | 145 | - |
| APTOS [25] | India | 3,662 | 1,805 | 1,857 | - |
| FCM-UNA [26] | Paraguay | 757 | 187 | 570 | - |
| Messidor-1 [27] | France | 1,200 | 546 | 654 | - |
| Messidor-2 [27,28] | France | 1,744 | 1,017 | 727 | - |

*Development dataset.*

The dataset used to develop the inherently interpretable deep learning model was obtained from the Kaggle Diabetic Retinopathy challenge [18] which initially contained records of 44,351 subjects with 88,702 retinal fundus images from both eyes (Fig 1A). This dataset was originally provided by EyePacs Inc., a diabetes screening program in California. A comparable dataset also obtained from EyePacs Inc. included ethnicity information and contained about 70% images from patients with Latin American ethnicity [29]. We automatically quality filtered the fundus images using an ensemble of 10 EfficientNets models [30] trained on the DeepDRiD dataset [31]. This model achieved a quality filtering accuracy of 87.5% [32]. After quality filtering, we retained 45,923 images from 28,984 subjects for training, with 73% of images in the healthy class and 27% in the DR class. The dataset was split into training, validation, and test folds with 75%, 10%, and 15% of images, respectively, making sure that all images from the same subject were allocated to the same fold. The training fold was used for model fitting, the validation fold for model selection and hyperparameter tuning, and the test fold for internal evaluation.

To evaluate the explanations provided by the explainable sparse BagNet model, three ophthalmologists (authors AR, LaK, and NS with 5, 9, and 14 years of experience respectively) marked the location of DR-related lesions on 65 randomly selected fundus images from the test set (20 grade 1 and 45 grade 2) using a custom-written annotation browser interface (S1 Fig) based on the Python web framework Django, version 4.2.1, with a secure PostgreSQL database, version 15.3, and a Javascript front-end (available at https://github.com/berenslab/retimgtools/releases/tag/v1.1.0). Annotators were asked to mark "Microaneurysms (MA)", "Hemorrhages (HE)", "Exudates (EX)", "Soft Exudates (SE)" or "Other" for lesions visible on the fundus image. We combined the annotations of all graders into a consensus annotation for each image (S1 Table). We also assessed the consistency between ophthalmologists' annotations by calculating the dice between their annotations, showing that annotating DR-related lesions exhaustively is a challenging task (S2 Table).

*External datasets.*

Additional fundus data sets were obtained from various sources (Table 1) and were used for external evaluation of the model to assess the generalization performance. In addition to

reference DR grades, some of these external datasets [19–23] contained pixel-wise annotations for disease-related lesions. We used these additional annotations to evaluate the performance of the interpretable deep-learning model at localizing DR-related lesions.

*Preprocessing.*

Raw fundus images were preprocessed by cropping them to a square size of 512 x 512 pixels using a circle fitting method [33]. Then, image intensities were normalized by the mean and standard deviation of the training set. We applied this preprocessing procedure to all the fundus images from all datasets with the same parameters.

## Inherently interpretable deep learning model for Diabetic Retinopathy detection

*Architecture.*

We trained and evaluated an inherently interpretable deep convolutional neural network (sparse BagNet [13,14]) for early DR detection. The sparse BagNet is an implicitly patch-based model based on bag-of-local features and aggregates local evidence from interpretable heatmaps to make predictions (Fig 1B). It takes a two-dimensional fundus image as input (Fig 1B.1) and outputs a binary prediction, which indicates the absence or presence of DR, together with the confidence as the probability score.

In contrast to other deep learning models, the sparse BagNet architecture is designed to be inherently interpretable, as the input image is implicitly split into many small, overlapping patches (size $q$ = 33x33 pixels corresponding to the size of the model's effective receptive field with stride $s$ = 8; Fig 1B.1), which are independently processed in parallel (Fig 1B.2) to compute the local evidence for the presence of DR. The patchwise predicted local evidence values are combined into a single class evidence map corresponding to a downsampled version of the input image (Fig 1B.3), which then is aggregated using average pooling and passed through a softmax function (Fig 1B.4) to output the probability distribution of DR (Fig 1B.5). Crucially, we employ a $\ell_1$-penalty on the local evidence to encourage a sparse class evidence map.

After inference, the model can support screening not only with the final prediction but also with the class evidence map (Fig 1B.3) highlighting the contribution of small local regions to the final prediction. To this end, the evidence map is upsampled to the full image resolution and overlaid on the input image. In contrast to post-hoc gradient-based methods [11], the class evidence map provided by the sparse BagNet is a transparent part of the actual decision-making process and faithfully captures the local evidence. We supplement the class evidence map by extracting patches from regions with high DR evidence (Fig 1B.5).

*Training procedure.*

We trained the model on the training set by minimising the following loss function including the $\ell_1$-penalty:

$$L\big((\mathbf{X}, \theta), \mathbf{y}\big) = CE\big(f(\mathbf{X}, \theta), \mathbf{y}\big) + \lambda \sum_{i,j,c} |\mathbf{A}_c^{ij}|. \tag{1}$$

Here, $\mathbf{X} \in \mathbb{R}^{H \times W \times C}$ denotes the input image with $H, W, C$ being height, width, and the number of channels, CE is the cross-entropy, $\mathbf{y}$ are the reference class labels, $f$ is the model with parameters $\theta$, and $\mathbf{A}_c$ denotes the evidence map of class $c$. The sparsity of the evidence maps depends on the hyperparameter $\lambda$.

We initialized the model with weights pre-trained on ImageNet and then retrained and optimized for accuracy on the Kaggle DR dataset for 100 epochs. We used the stochastic gradient descent optimizer with an initial learning rate of $10^{-3}$, and a clipped cosine learning rate scheduler with a minimum value set to $10^{-4}$. We performed data augmentation during training by applying random cropping, flipping, color jitter, translation, and rotation following [34]. The sparsity hyperparameter $\lambda$ was chosen based on the classification accuracy on the validation set (S2 Fig).

## Baseline model and post-hoc interpretability

For comparison, we trained a standard black-box ResNet-50 [35] for early onset DR detection using the same training procedure as described above. We evaluated the classical interpretability techniques Integrated Gradients and Guided Backpropagation due to their high performance in identifying clinically validated DR lesions [36].

## Clinical user study for AI-based decision support

*Study dataset.*

The user study was designed to evaluate the usefulness of the explanations provided by the inherently interpretable deep learning model in clinical practice. The dataset for each grading task (see below) consisted of 60 fundus images from the internal test set, where 20 images were sampled from grade 0, grade 1, and grade 2 respectively. For each grade, 15 images were correctly classified by the network and 5 falsely, making this a challenging screening task for clinicians. Thus, the fraction of images with DR in the user study was 66% and the deep learning model achieved an accuracy of 75% by design. Image grading was based solely on the fundus image and AI support, but no additional clinical data were provided.

*Study design.*

Six trained ophthalmologists with a median clinical experience of 9 years (4–17 years) participated in the reader study (including authors LaK, AR, and NS). We did not perform a formal power calculation. The study consisted of three tasks: In task 1 (referred to as "H"), participants were asked to grade fundus images without AI support (S3 Fig). In task 2 ("H+AI"), participants were additionally provided with the class predicted by the deep learning model and its confidence (S4 Fig). Finally, in task 3 ("H+XAI"), participants were additionally shown model explanations in the form of up to 12 bounding boxes around the regions from the class evidence map with the highest evidence, with bounding boxes matching the effective receptive field size and depicting the local image patches that contribute most to the global class evidence (S5 Fig).

For the three grading tasks, readers were instructed to classify each fundus image into two classes ("No DR" and "DR"). They were told to classify an image as "DR" even if they thought it only contained signs of mild non-proliferative DR (grade 1). None of the readers had access to the true labels. For task 3, readers were told that some bounding box explanations may contain healthy regions, as the algorithm also generated bounding boxes for healthy images erroneously classified as DR by the sparse BagNet model. In addition to the assigned class, we recorded the time it took for the reader to grade each image and asked them to rate their confidence on a scale from 1 to 5. Ethical approval for the study was obtained from the Ethics Committee at the University Hospital Tübingen (Ref No. 249/2023BO2).

A custom-written browser interface based on the Python web framework Django (version 4.2.1) with a secure PostgreSQL database (version 15.3) and a JavaScript front-end

was used to carry out the study (S3 Fig, S4 Fig, S5 Fig). The tool showed the fundus image, and response options and provided a digital magnifier to enlarge small image regions.

## Evaluation criteria and statistical analysis

Criteria for evaluating the performance of the inherently interpretable deep learning model were specified before the start of the study based on previous work [13]. We evaluated three aspects of the model's quality:

1. DR screening performance compared to a regular deep learning model, within and across datasets.
2. The quality of the class evidence maps and derived bounding boxes in terms of lesion localization.
3. The usefulness of the inherently interpretable deep-learning model and the derived bounding boxes for decision support.

*DR screening performance.*

The primary measure of DR screening performance was the accuracy of the model for early DR detection using the reference labels. Additionally, we evaluated the area under the receiver-operating curve (AUC), sensitivity, specificity, and precision. All measures were computed on the internal test set as well as on the ten external datasets (Table 1). The model was not retrained or fine-tuned before assessment on the external datasets. All measures were computed using the scikit-learn package (v 1.0.2) and confidence intervals were computed using a bootstrap procedure with 1000 unstratified resamples [37].

*Quality of class evidence maps.*

To measure the quality of the class evidence maps and the derived bounding boxes for lesion localization, we calculated the proportion of highlighted regions (regions within the bounding box) that contained annotated lesions ("localization precision"). To this end, we used the annotations collected for this study on 65 images from the test set, as well as those external datasets containing pixel-level annotations (Table 1). We did not evaluate the fraction of lesions detected by our model ("recall"), as we did not train the model for lesion detection, and diagnostic support does not require an exhaustive detection of all lesions.

*Statistical analysis of decision support.*

We measured the performance of the readers in our clinical user study as the accuracy of the reader's decision with respect to the reference labels. To assess the effect of the task and DR reference grade statistically, we fit the responses with a generalized linear model (R, function *glm*, v 4.0.3) with predictor *task* or with predictors *task* and *DR grade* including interactions. If we found significant predictors at the $\alpha$ = 0.05 level, we computed the marginal means and 95%-confidence intervals (package *emmeans*, v 1.5.3) as well as the respective contrasts between conditions for post-hoc testing. Tukey's method was used for correcting for multiple comparisons. We used the same procedure for analyzing the measured grading time and the reported confidence, but used a linear model (function *lm*) instead.

## Role of the funding source

The funders of this work had no role in the study design, collection, analysis, and interpretation of data, the writing of the report, nor in the decision to submit the paper for publication.

## Results

We trained and evaluated an inherently interpretable deep learning model ("sparse Bag-Net") for early DR screening (Fig 1B). We first evaluated screening performance for early DR against the state-of-the-art non-interpretable black-box model ("ResNet50") on the internal test set of the development dataset and on a large number of additional datasets (see Table 2). The sparse BagNet performed well and was comparable to the state-of-the-art model on the internal test set (accuracy: 0.906, 95% CI [0.900–0.913]; AUC: 0.904 [0.894–0.913]; sensitivity: 0.709 [0.688–0.729]; specificity: 0.977 [0.973–0.981]; precision: 0.918 [0.903–0.932]) and generalized well to a number of external datasets (Table 2).

The key advantage of our inherently interpretable model is that the local disease evidence is explicitly represented in a class evidence map (Fig 1B.3 and Fig 2B). During training, the class evidence map is encouraged to be sparse, such that the final loss function balances prediction accuracy and an interpretable map. For the model studied above, the regularization parameter trading-off accuracy and sparseness was heuristically chosen such that sparseness was encouraged at a minimal loss of accuracy (S2 Fig). At each location in the class activation map, the color indicates the model output for an individual image patch. We detected the regions with the highest evidence and placed bounding boxes corresponding to the patch size around these points (Fig 2A).

**Table 2. Summary of the classification performance with confidence intervals (CIs) computed at 95% using bootstrapping (n=1000). "AUC" refer to the receiver-operating curve. "Loc Bag" and "Loc GBP" respectively refer to the localization precision of the sparse BagNet and Guided Backpropagation on ResNet-50 at localizing lesions from annotated images. For each dataset, the first row shows the performance of the interpretable sparse BagNet model, while the second row shows the performance of the baseline black-box ResNet-50 model. The Kaggle dataset (first row) is the internal dataset used to train and evaluate the model, while the other datasets were used for external validation to assess the generalization properties of the trained model. The low classification performance on the FCM-UNA and FGA-DR datasets can be explained by the relatively low quality of most images in the FCM-UNA dataset and the large intensity variation of the FGA-DR dataset (S6 Fig). The low localization precision (0.664) on the E-Ophtha dataset is likely due to annotations only being provided for "Microaneurysms" and "Exudate" lesions, while the images could contain other DR-related lesions.**

| Dataset | | Accuracy | AUC | Sensitivity | Specificity | Precision | Loc $_{Bag}$ | Loc $_{GBP}$ |
|---|---|---|---|---|---|---|---|---|
| Kaggle | Bag. | .906 (.900 - .913) | .904 (.894 - .913) | .709 (.688 - .729) | .977 (.973 - .981) | .918 (.903 - .932) | .941 | - |
| | Res. | .914 (.907 - .921) | .935 (.927 - .943) | .765 (.745 - .784) | .967 (.962 - .972) | .894 (.878 - .908) | - | .656 |
| | | .891 (.864 - .917) | .879 (.838 - .913) | .951 (.927 - .972) | .768 (.699 - .828) | .895 (.861 - .925) | .804 | - |
| IDRiD | | .882 (.851 - .909) | .864 (.822 - .902) | .963 (.942 - .981) | .714 (.639 - .781) | .875 (.84 - .908) | - | .140 |
| | | .903 (.864 - .917) | .944 (.838 - .913) | .920 (.927 - .972) | .892 (.699 - .828) | .851 (.861 - .925) | .656 | - |
| E-Ophtha | | .933 (.851 - .909) | .972 (.822 - .902) | .966 (.942 - .981) | .912 (.639 - .781) | .880 (.840 - .908) | - | .030 |
| | | .799 (.781 - .819) | .789 (.752 - .823) | .811 (.793 - .830) | .594 (.500 - .687) | .972 (.963 - .980) | .872 | - |
| FGA-DR | | .763 (.743 - .781) | .816 (.768 - .858) | .764 (.743 - .783) | .743 (.653 - .819) | .981 (.973 - .987) | - | .336 |
| | | .831 (.753 - .899) | .931 (.870 - .981) | .821 (.733 - .898) | 1 | 1 | .881 | - |
| DIARETDB1 | | .742 (.652 - .831) | .811 (.715 - .900) | .738 (.640 - .829) | .800 (.333 - 1.00) | .984 (.950 - 1.00) | - | .000 |
| | | .825 (.818 - .832) | .926 (.922 - .931) | .669 (.657 - .681) | .980 (.977 - .984 | .971 (.966 - .976) | .965 | - |
| DDR | | .887 (.881 - .892) | .963 (.960 - .966) | .800 (.790 - .810) | .973 (.968 - .977) | .967 (.962 - .972) | - | .249 |
| | | .879 (.847 - .908) | .922 (.889 - .951) | .662 (.584 - .742) | .983 (.968 - .997) | .950 (.905 - .990) | - | |
| DR2 | | .876 (.845 - .906) | .866 (.825 - .905) | .669 (.591 - .742) | .977 (.959 - .993) | .933 (.884 - .975) | - | |
| | | .973 (.968 - .979) | .995 (.992 - .996) | .982 (.975 - .987) | .965 (.956 - .973) | .966 (.958 - .974) | - | |
| APTOS | | .949 (.942 - .956) | .972 (.965 - .978) | .942 (.931 - .952) | .956 (.946 - .965) | .956 (.947 - .966) | - | |
| | | .773 (.744 - .802) | .936 (.918 - .952) | .702 (.664 - .738) | .989 (.972 - 1.00) | .995 (.987 - 1.00) | - | |
| FCM-UNA | | .877 (.853 - .900) | .967 (.954 - .979) | .840 (.811 - .868) | .989(.971 - 1.00) | .996 (.989 - 1.00) | - | |
| | | .889 (.871 - .907) | .943 (.929 - .955) | .832 (.804 - .859) | .958 (.939 - .974) | .959 (.941 - .975) | - | |
| Messidor-1 | | .893 (.876 - .909) | .954 (.942 - .965) | .852 (.823 - .878) | .943 (.923 - .963) | .947 (.928 - .964) | - | |
| | | .829 (.812 - .847) | .876 (.859 - .894) | .750 (.719 - .785) | .886 (.865 - .906) | .825 (.794 - .853) | - | |
| Messidor-2 | | .851 (.835 - .869) | .925 (.912 - .938) | .794 (.763 - .823) | .893 (.875 - .913) | .841 (.815 - .868) | - | |

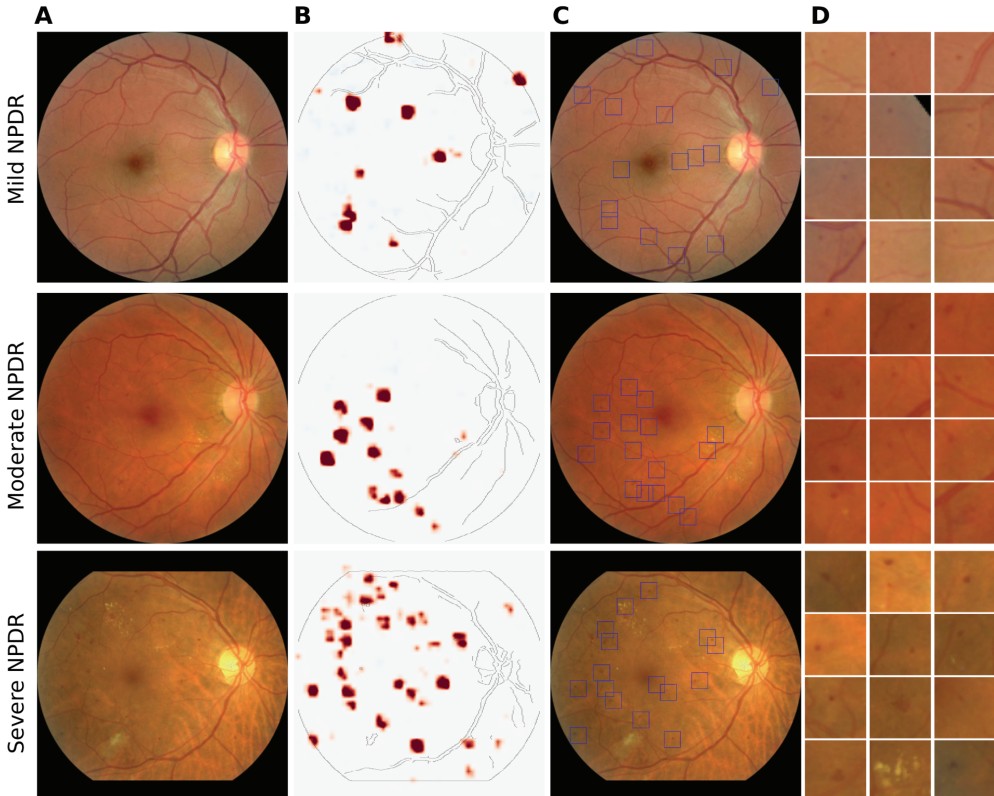

**Fig 2. Inherently interpretable deep learning framework highlights clinically relevant image regions.** (A) Examples of retinal fundus images from different DR grades (top to bottom: mild NPDR, moderate NPDR and severe NPDR). (B) Class evidence map extracted from the inherently interpretable model without further processing. Red regions indicate evidence for the presence of at least mild DR. (C) Bounding boxes drawn around suspicious regions in the class evidence map. In some cases, the bounding boxes are placed in regions for which there is no visible evidence due to the scaling of the color map. Yet, these evidence values are also strictly positive. (D) Suspicious regions from (C) enlarged and sorted with decreasing evidence scores. Depending on the image grade, the suspicious regions contain various DR-related lesions such as microaneurysms, hemorrhages, or drusen.

Although the model was never trained with pixel-level annotations or supervision signals other than the image-level DR reference label, the highlighted regions typically contained DR-related lesions such as microaneurysms, drusen, or hemorrhage with high precision (Fig 3).

We quantitatively evaluated how well the class evidence maps provided information about the location of disease-related lesions using a subset of images from the test set of the development dataset (Fig 3) as well as external datasets with pixel-level annotations (Table 1). The class evidence maps precisely localized DR lesions, as most regions flagged as suspicious indeed contained annotated lesions (Table 2, last column). For the images from the development dataset, we obtained a precision of 0.960 (95% CI [0.941–0.976]), with minor differences between images with mild and moderate NPDR (0.783 vs. 0.970). Notably, our model generalized well to external test sets, with precision ranging from 0.656 to 0.965 (Table 2, last column).

We also evaluated suspicious regions extracted from images the algorithm falsely classified as DR with high confidence (>0.75). To this end, we showed two clinicians 30 images falsely classified as DR with bounding boxes (S8 Fig). Sometimes, these image patches showed

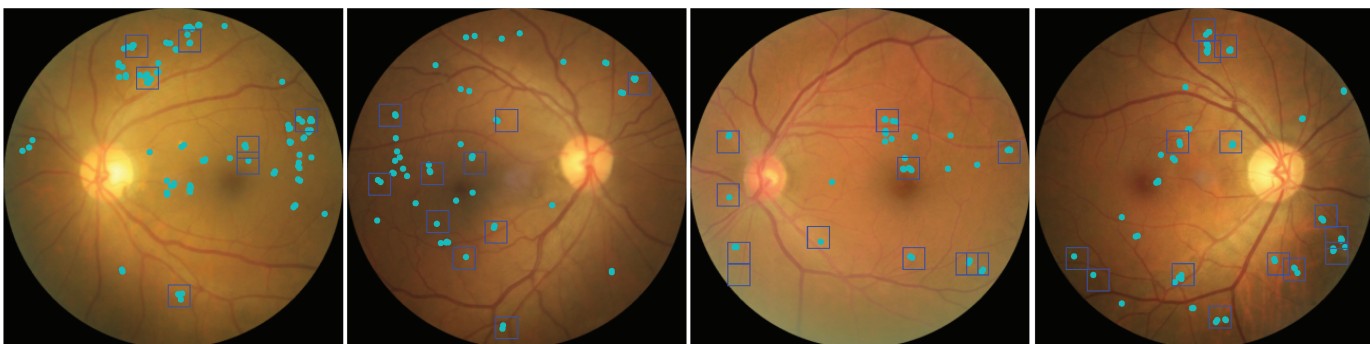

**Fig 3. Extracted high evidence images patches contain DR-related lesions.** Example fundus images with DR, with DR lesions identified by three clinicians (cyan). Bounding boxes (blue) were extracted from the class evidence maps based on regions of high evidence for DR. Note that all bounding boxes contain annotated lesions, but – as the number of bounding boxes per image was restricted to twelve – not all lesions are contained in bounding boxes.

unclear or ambiguous lesions unrelated to DR, but they typically contained anomalies related to DR such as microaneurysms or exudates, but not in a number or severity sufficient for clinical DR diagnosis (S8 Fig).

We next compared the localization performance of the inherently interpretable sparse BagNet to classic post-hoc methods such as Integrated Gradients [38] or Guided Backprop applied to the state-of-the-art model (Fig 4A–4C). These methods were chosen because they performed well in a clinical validation of post-hoc explainability techniques for DR [36]. We found that bounding boxes obtained from Guided Backprop or Integrated Gradients were much less precise in localizing DR-related lesions (0.941 vs. 0.656, Fig 4D, Table 2), especially for out-of-sample test datasets.

We then investigated whether our interpretable deep learning model could effectively aid clinicians in detecting DR via a retrospective reader study with six experienced ophthalmologists screening fundus images for the presence of early DR with various levels of AI assistance (see Methods). Without AI assistance (labeled "H") ophthalmologists reached a mean classification accuracy of 0.611 (95% CI [0.560–0.660]; Fig 5A). Their accuracy increased significantly to 0.758 ([0.711–0.800], $p = 0.0001$, post-hoc test with Tukey's correction for multiple comparisons, see Methods) when they had access to the deep learning model's prediction and confidence ("H+AI"). They achieved similar performance with additional access to AI explanations in the form of bounding boxes around suspicious regions extracted from the class evidence maps ("H+XAI") at an accuracy of 0.786 [0.741–0.825].

We studied ophthalmologists' performance in screening for DR in fundus images of different disease grades in more detail (Fig 5B). Without AI support, detecting images with mild DR (grade 1) was the most challenging with comparably low performance, which improved with AI support. For healthy images, screening performance improved significantly with any form of AI decision support (H: 0.567, [0.477–0.652]; H+AI: 0.842, [0.765–0.897]; H+XAI: 0.817, [0.737–0.876]; H vs. H+AI: $p < 0.0001$; H vs. H+XAI: $p = 0.0001$; H+AI vs. H+XAI: $p = 0.8645$), while for images with mild DR, we observed that screening only improved significantly for AI support with explanations (H: 0.483, [0.395–0.572]; H+AI: 0.617, [0.527–0.699]; H+XAI: 0.733, [0.647–0.805]; H vs. H+AI: $p = 0.0962$; H vs. H+XAI: $p = 0.0003$; H+AI vs. H+XAI: $p = 0.1326$). For images with moderate DR, AI support had no significant effect on screening performance. Taken together, this provides evidence that giving ophthalmologists access to AI support led to superior DR screening performance, with explanations based on the sparse BagNet model being most effective for difficult diagnostic decisions.

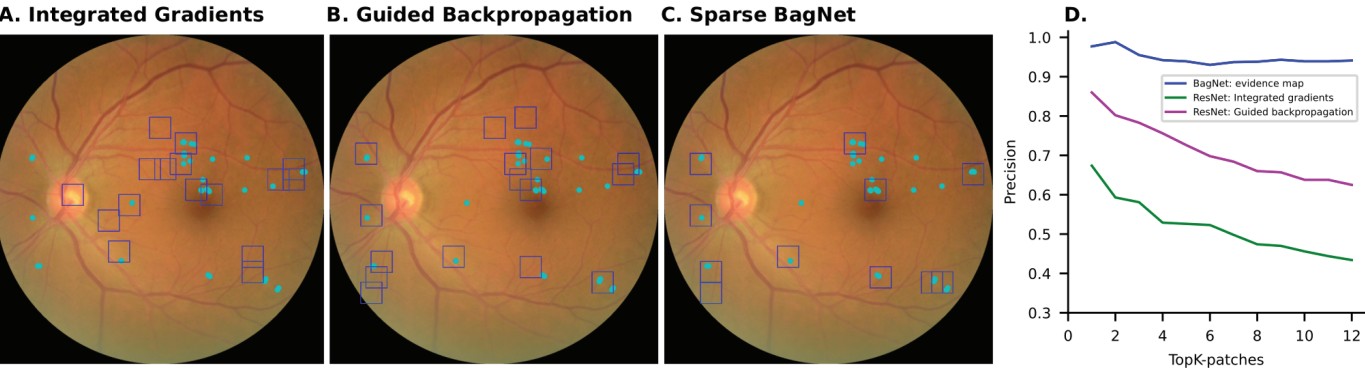

**Fig 4. Inherently interpretable deep learning framework highlights lesions more precisely than post-hoc techniques applied to a standard DNN.** (A) Suspicious regions (blue) marked with bounding boxes extracted from the heatmap obtained with Integrated gradients from the standard DNN. Clinically relevant DR lesions are marked in cyan. (B) As in (A) extracted from the heatmap obtained with Guided backpropagation. (C) For comparison, suspicious regions were obtained from the SparseBagNet. (D) Systematic comparison of localization precision for clinically annotated DR lesions as a function of the number of considered patches.

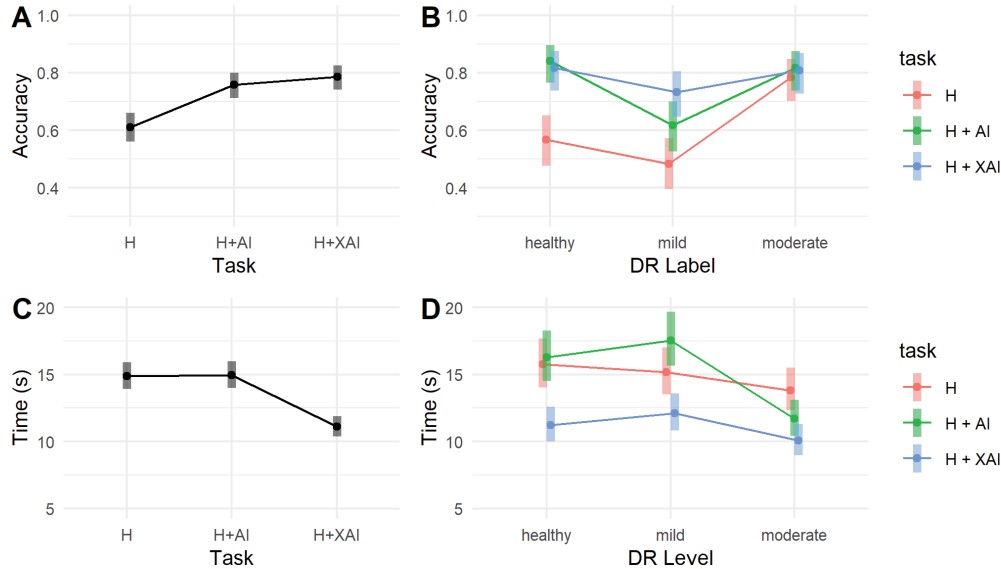

**Fig 5. Providing AI-based clinical decision support based on the inherently interpretable deep learning model improves DR screening.** (A) Screening accuracy with different levels of AI assistance. Six ophthalmologists graded fundus images without AI assistance ("H"), with access to the AI prediction ("H+AI"), and with additional access to AI explanations ("H+XAI"). AI assistance improved screening accuracy, but access to AI explanations had only a small additional effect. (B) Screening accuracy for DR on fundus images of different disease grades. For healthy images, accuracy improved significantly with any form of AI decision support ("H+AI" or "H+XAI"), while for images with mild DR, screening improved significantly for AI support with explanation ("H+XAI"). For images with moderate DR, AI support had no significant effect on screening performance. (C) Screening time in screening DR with different levels of AI assistance. The decision time is significantly reduced with AI support ("H+XAI") with explanation compared to the other tasks ("H", and "H+AI"). (D) Screening time in screening for DR on fundus images of different disease grades. Screening time reduces at all disease stages with a significant effect of AI decision support with explanation for healthy images ("grade 0"), mild DR ("grade 1"), and moderate DR ("grade 2").

We next studied whether AI decision support would not only allow ophthalmologists to make more accurate screening decisions but also reach their decisions faster. We found that

the decision time was significantly reduced when providing ophthalmologists AI support with explanations compared to both other tasks (Fig 5A, H: 15.2 s [14.1-16.4]; H+AI: 15.9 s [14.7-17.1]; H+XAI: 11.7 s [10.8-12.6]; H vs. H+AI: $p = 0.7435$; H vs. H+XAI: $p < 0.0001$; H+AI vs. H+XAI: $p < 0.0001$). This reduction was present at all disease stages, with a significant effect of AI decision support with explanations for healthy images (Fig 5A; H: 15.8 s [14.1-17.7]; H+AI: 16.3 s [14.5-18.3]; H+XAI: 11.2 s [10.0-12.6], H vs. H+AI: $p = 0.9153$; H vs. H+XAI: $p < 0.0001$; H+AI vs. H+XAI: $p < 0.0001$), mild DR (H: 15.2 s [13.5-17.0]; H+AI: 17.5 s [15.6-19.7]; H+XAI: 12.1 s [10.8-13.6], H vs. H+AI: $p = 0.1843$; H vs. H+XAI: $p = 0.180$; H+AI vs. H+XAI: $p < 0.0001$), as well as moderate DR (H: 13.8 s [12.3-15.5]; H+AI: 11.7 s [10.4-13.1]; H+XAI: 10.1 s [9.0-11.3]; H vs. H+AI: $p = 0.1058$; H vs. H+XAI: $p = 0.004$; H+AI vs. H+XAI: $p = 0.1724$). In summary, this indicates that decision support with accurate explanations provided by the sparse BagNet model could reduce screening times across all disease levels.

We also analyzed whether AI decision support would change the confidence with which the ophthalmologists could grade the images, but did not find a significant effect of AI support (H: 3.8 [3.7-3.9]; H+AI: 3.7 [3.6-3.9]; H+XAI: 3.6 [3.5-3.7], H vs. H+AI: $p = 0.6806$; H vs. H+XAI: $p = 0.0543$; H+AI vs. H+XAI: $p = 0.3023$). We conclude that self-reported confidence may not be a reliable measure of grader uncertainty compared to recorded decision time.

We finally analyzed whether the positive effect on accuracy was dependent on whether the deep learning model had classified the image correctly or not, as AI support has been reported to be detrimental in case of model errors [39]. In line with the results above, we found that screening performance and decision time significantly improved for cases in which the deep learning model had made a correct decision (S5 Fig; accuracy, H vs. H+AI: $p<0.0001$; H vs. H+XAI: $p<0.0001$; H+AI vs. H+XAI: $p<0.0001$; time, H vs. H+AI: $p = 0.8178$; H vs. H+XAI: $p<0.0001$; H+AI vs. H+XAI: $p<0.0001$). For cases in which the model had made an incorrect decision, we neither detected positive nor negative effects on accuracy (H vs. H+AI: $p<0.3216$; H vs. H+XAI: $p = 0.4953$; H+AI vs. H+XAI: $p = 0.9480$) and slightly positive effects on decision time (H vs. H+AI: $p = 0.4557$; H vs. H+XAI: $p = 0.0941$; H+AI vs. H+XAI: $p = 0.0031$) meaning that the decision time was still smaller despite the wrong prediction of the model.

## Discussion

In this study, we trained and evaluated an inherently interpretable deep-learning model for early diabetic retinopathy detection. This is a challenging task even for experienced ophthalmologists. Our model achieved a classification performance comparable to the black-box baseline model in the internal test set and on ten publicly available external datasets. While the training dataset contained a large fraction of images from patients of Latin American ethnicity, the external datasets were acquired in diverse world regions and different devices, thus that our model showed a good generalization across different ethnic groups and patient populations. While some of these datasets also contained patients of African ancestry, none of the datasets were acquired on the African continent.

In addition to a binary diagnostic decision that is commonly communicated in DR screening settings, our model provides explanations via interpretable evidence maps, which highlight regions of the image used by the network in making its decisions. We found that the inherently interpretable framework precisely located disease-related lesions in the image,

more so than post-hoc techniques applied to a state-of-the-art DNN, in particular for out-of-sample test datasets. Even in case of incorrect model predictions according to the reference labels, model explanations proofed to be useful and highlighted suspicious regions.

In a retrospective reader study, we found that highlighting these regions during grading helped ophthalmologists improve their grading performance, especially for difficult cases, while reducing their decision time. This indicates that current paradigms used in AI-based screening scenarios may benefit from including explanations for easier human verification and enhanced trust in the algorithms decision [3,5]. Our study further showed that the errors of the AI model did not negatively affect decision-making by ophthalmologists, in contrast to earlier human-AI studies on clinical decision support [39,40]. A limitation of our model is that it was trained on a dataset from North America, and may need to be fine-tuned on data from the intended target population, although its generalization results on ten additional datasets were promising.

As the potential of AI for medical image analysis has become evident [41,42], such systems have reached performance close to, or even superior to, those of clinical experts in a variety of tasks [43]. More recently, the focus has shifted towards AI systems assisting clinicians in making better decisions [39]. In this setting, clinicians need to understand how decisions are formed by the AI model, such that transparency and interpretability of medical AI systems have become important aspects [7,11,12,44]. In agreement, the need for trustworthy and transparent AI systems and effective human/AI collaboration has been identified in standardized guidelines to facilitate their adoption in clinical practice [44]. While this generally poses challenges in balancing high performance and interpretability [45], our study has shown that inherent interpretability can be achieved without significant performance trade-offs if the inductive biases of the interpretable model are met – in our case, as early DR causes only very localized lesions in the retina. Other inherently interpretable models include prototype-based networks [46], which are difficult to use for diseases with many small, distributed lesions, for which the training procedure is more complex and for which interpretability is not straightforward [47].

In a clinical setting, such an inherently interpretable model could assist clinicians in mitigating the challenge of early and accurate diagnosis of presymptomatic diseases, such as diabetic retinopathy detection. Given several approved AI systems for DR screening, clinical implementation could be comparatively straightforward. The trained model can efficiently generate predictions, on a time scale not impeding on clinical practice ($\ll 1s$/image), requires relatively little memory ($\sim 350mb$), and does not require additional models to run to create explanations. Such explanations could be added to existing reporting templates in commercial AI systems, allowing screeners to quickly ascertain the plausibility of the models prediction. In this setting, real-world prospective studies could be conducted to test the impact of the explanations obtained from our model on screening quality and speed, in particular for patient with beginning DR.

One limitation of our model is that it may not provide good explanations if its inductive bias is not matched to the disease, e.g. when lesions cover large parts of the retina as in more advanced DR grades [13]. Future applications also include time-to-progression prediction for diseases like DR [48] through interpretable-by-design deep survival models [49].

## Supporting information

**S1 Fig**. **Web interface for the annotation task.** A fundus image is shown and based on it, the annotator is asked to annotate lesions related to Diabetic Retinopathy. By moving the mouse over a region of the image, an enlarged version of that region is displayed. All images are from

patients with DR of grade 1 ("mild DR") or 2 ("moderate DR"). Each lesion is marked by selecting the type (Microaneurysms: MA, hemorrhages: HE, exudates: EX, soft exudate: SE, artifact, or any other lesions) and clicking on the image location.
(TIF)

**S1 Table**. **Summary of model performance on localizing DR-related lesions from graders' annotations** The precision of the model on each clinician annotation is calculated as the proportion of bounding boxes from regions highlighted on heatmaps containing lesions annotated by a grader. The random precision is obtained by drawing 20 random bounding boxes over each annotated image, excluding those falling in regions containing more than 10% black pixels. The union "∪" gives the precision of the model with the combined clinicians' annotation masks, while the intersection "∩" gives the precision of the model with reference annotation masks obtained as the intersections of clinicians' annotation over each image.
(PDF)

**S2 Table**. **Inter-grader performance on 65 fundus images from the internal Kaggle test set annotated by three ophthalmologists** "Grader X - Grader Y" refers to the dice score between grader X and grader Y. The Dice score is calculated for each pair of graders as the overlap between their annotation using a patch size of 33×33 pixels corresponding to the receptive field of the model and considering different strides (s = 8, 32 for overlapping patches and s=33 for non-overlapping patches). "Grader X - Grader Y ∪ Grader Z" refers to the dice score between grader X, Y, and Z while "Grader Y ∪ Grader Z" is the union between grader Y and Z, and "Grader Y ∩ Grader Z" is the intersection between grader Y and Z.
(PDF)

**S2 Fig**. **Comparison of the sparse BagNet performance with different regularization values on the validation dataset** The regularization coefficient $\lambda$ affects the classification performance (accuracy and AUC) of the model. The red points indicate the selected value, which is a compromise between sparsity and both accuracy and AUC. It also defines the trade-off between the model's interpretability and classification performance.
(TIF)

**S3 Fig**. **Web interface for the grading task without AI support ("H")** A fundus image is shown and based on it, the grader is asked to decide whether the corresponding patient has Diabetic Retinopathy (DR) of any severity, including mild DR. In addition, the grader is asked to rate the confidence of his/her decision on a scale from 1 (least confident) to 5 (most confident). By moving the mouse over a region of the image, an enlarged version of that region is displayed. The time taken to reach each decision (grading and confidence) is recorded.
(TIF)

**S4 Fig**. **Web interface for the grading task with AI support ("H + AI")** A fundus image is shown with the model's prediction and its confidence level (from 0% to 100 %, with 100% being the highest confidence score). Based on this, the grader is asked to decide whether the corresponding patient has Diabetic Retinopathy (DR) of any severity, including mild DR. In addition, the grader is asked to rate the confidence of his/her decision on a scale from 1 (least confident) to 5 (most confident). By moving the mouse over a region of the image, an enlarged version of that region is displayed. The time taken to reach each decision (grading and confidence) is recorded.
(TIF)

**S5 Fig**. **Web interface for the grading task with AI support and explanations ("H + XAI").** A fundus image is shown with the model's prediction, its confidence level (from 0% to 100 %,

with 100% being the highest confidence score), and explanation in the form of blue bounding boxes around the regions for which the AI model believes that they contain signs of DR. Based on this, the grader is asked to decide whether the corresponding patient has Diabetic Retinopathy (DR) of any severity, including mild DR. In addition, the grader is asked to rate the confidence of his/her decision on a scale from 1 (least confident) to 5 (most confident). By moving the mouse over a region of the image, an enlarged version of that region is displayed. The time taken to reach each decision (grading and confidence) is recorded.
(TIF)

**S6 Fig**. **Examples of fundus images from each dataset.**
(TIF)

**S7 Fig**. **Heatmap with combined clinicians' annotations of four examples of fundus cases with DR.** For each example, the left side shows the heatmap with clinicians' annotations and bounding boxes around the regions of positive activation, while the right side shows the fundus image with clinicians' annotations and bounding boxes around the regions of positive activation.
(TIF)

**S8 Fig**. **Examples of high-confidence false positives analyzed by two clinicians.** On the left side of each example, the image displays bounding boxes highlighting regions with positive activation. On the right side, the suspicious regions from the left are enlarged and arranged in descending order of evidence scores. (A) A false-positive image where the clinicians interpreted the suspicious regions as "vitreous opacities" and "uveitis vitreous cells," respectively. (B) A false-positive image where one clinician identified the suspicious regions as "synchisis scintillans," while the other suggested the patient may have recently received an intravitreal steroid injection. (C) A false-positive image where both clinicians identified the suspicious regions as "microaneurysms" possibly associated with bleeding. (D) A false-positive image where both clinicians recognized the suspicious regions as "microaneurysms" and "hard exudates". **(E, F)** False-positive images where one clinician classified the image as DR while the other classifies it as no DR, citing the presence of only a single microaneurysm lesion in the suspicious regions.
(TIF)

**S9 Fig**. **Analysis of errors of the AI model on accuracy and decision times for different tasks during the retrospective reader study. (a)** For all tasks, ophthalmologists' accuracy is higher when the deep learning model makes the correct decision. For correct classifications, the AI assistance improves grading accuracy. For incorrect classification, it does not make it worse. **(b)** Ophthalmologists' decision time decreases overall when the deep learning model makes the correct decision. When the AI model is correct, the explanation decreases decision time significantly, while it does not increase the decision time for incorrect decisions.
(TIF)

## Acknowledgments

We thank Sarah Müller, Pearse Keane and Murat Ayhan for discussion and Murat Ayhan quality filtering code.

## Author contributions

**Conceptualization:** Kerol Djoumessi, Lisa M. Koch, Philipp Berens.

**Data curation:** Kerol Djoumessi, Ziwei Huang, Annekatrin Rickmann, Natalia Simon, Lisa M. Koch.

**Formal analysis:** Kerol Djoumessi.

**Funding acquisition:** Philipp Berens, Lisa M. Koch.

**Investigation:** Kerol Djoumessi, Laura Kühlewein, Lisa M. Koch, Philipp Berens.

**Methodology:** Kerol Djoumessi, Philipp Berens.

**Project administration:** Philipp Berens.

**Software:** Kerol Djoumessi, Ziwei Huang.

**Supervision:** Lisa M. Koch, Philipp Berens.

**Validation:** Laura Kühlewein, Annekatrin Rickmann, Natalia Simon.

**Visualization:** Kerol Djoumessi.

**Writing – original draft:** Kerol Djoumessi, Lisa M. Koch, Philipp Berens.

**Writing – review & editing:** Laura Kühlewein, Annekatrin Rickmann, Natalia Simon.

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
