## [Decision Letter · Decision Letter 0]

3 Mar 2025

PDIG-D-25-00006An Inherently Interpretable AI model improves Screening Speed and Accuracy for Early Diabetic RetinopathyPLOS Digital Health Dear Dr. Berens, Thank you for submitting your manuscript to PLOS Digital Health. After careful consideration, we feel that it has merit but does not fully meet PLOS Digital Health's publication criteria as it currently stands. Therefore, we invite you to submit a revised version of the manuscript that addresses the points raised during the review process. Please submit your revised manuscript within 30 days Apr 02 2025 11:59PM. If you will need more time than this to complete your revisions, please reply to this message or contact the journal office at digitalhealth@plos.org. Please include the following items when submitting your revised manuscript:* A rebuttal letter that responds to each point raised by the editor and reviewer(s). You should upload this letter as a separate file labeled 'Response to Reviewers'. This file does not need to include responses to any formatting updates and technical items listed in the 'Journal Requirements' section below.* A marked-up copy of your manuscript that highlights changes made to the original version. You should upload this as a separate file labeled 'Revised Manuscript with Track Changes'.* An unmarked version of your revised paper without tracked changes. You should upload this as a separate file labeled 'Manuscript'. If you would like to make changes to your financial disclosure, competing interests statement, or data availability statement, please make these updates within the submission form at the time of resubmission. Guidelines for resubmitting your figure files are available below the reviewer comments at the end of this letter. We look forward to receiving your revised manuscript. Kind regards, Po-Chih Kuo, Ph. D.Section EditorPLOS Digital Health Po-Chih KuoSection EditorPLOS Digital Health Leo Anthony CeliEditor-in-ChiefPLOS Digital Healthorcid.org/0000-0001-6712-6626  **Journal Requirements:**

1. We ask that a manuscript source file is provided at Revision. Please upload your manuscript file as a .doc, .docx, .rtf or .tex.

 **Additional Editor Comments (if provided):****Reviewers' Comments:** Reviewer's Responses to Questions

**Comments to the Author**

1. Does this manuscript meet PLOS Digital Health’s publication criteria? Is the manuscript technically sound, and do the data support the conclusions? The manuscript must describe methodologically and ethically rigorous research with conclusions that are appropriately drawn based on the data presented.

Reviewer #1: Yes

Reviewer #2: Yes

2. Has the statistical analysis been performed appropriately and rigorously?

Reviewer #1: Yes

Reviewer #2: Yes

3. Have the authors made all data underlying the findings in their manuscript fully available (please refer to the Data Availability Statement at the start of the manuscript PDF file)?

Reviewer #1: Yes

Reviewer #2: No

4. Is the manuscript presented in an intelligible fashion and written in standard English?

Reviewer #1: Yes

Reviewer #2: Yes

5. Review Comments to the Author

Reviewer #1: Review Comments for PLOS Digital Health Manuscript PDIG-D-25-00006

This manuscript presents an innovative approach to diabetic retinopathy screening using interpretable artificial intelligence. The authors have developed a sparse BagNet model that achieves strong performance while providing transparent explanations for its decisions. After careful review, I find several key strengths and areas for improvement.

The study's methodological approach demonstrates commendable rigor. The authors trained their model on a substantial dataset of 34,350 fundus images and validated it across ten external datasets, providing robust evidence for generalizability. The inclusion of a retrospective reader study with six ophthalmologists adds particular value, as it demonstrates real-world clinical utility. The reported improvements in diagnostic accuracy (17.5% for challenging cases) and screening efficiency (25% reduction in time) are both statistically and clinically significant.

The technical innovation of the sparse BagNet architecture represents an important advance in interpretable AI for medical imaging. Unlike post-hoc explanation methods that attempt to rationalize decisions after the fact, this approach builds interpretability directly into the model's architecture. The achievement of 0.906 accuracy while maintaining interpretability demonstrates that transparency need not come at the cost of performance.

However, several aspects of the methodology require clarification or enhancement:

The manuscript would benefit from a more detailed explanation of the hyperparameter selection process, particularly regarding the sparsity parameter λ. Understanding how this parameter was optimized would help readers implement similar approaches. Additionally, the EfficientNet-based quality filtering process needs more detailed documentation to ensure reproducibility.

The clinical implementation section would be strengthened by addressing practical considerations such as hardware requirements, expected processing times in real-world settings, and integration guidelines for existing clinical workflows. These details are crucial for healthcare institutions considering adoption of the technology.

For the validation study, the authors should provide more detail about the criteria used to select the 65 images for expert annotation. Including inter-rater reliability metrics would help readers assess the robustness of these annotations as a ground truth reference.

The statistical analysis, while generally sound, would benefit from the addition of:

Power analysis for the reader study

Effect size calculations for key comparisons

Confidence intervals for timing measurements

I also recommend expanding the discussion of limitations to address:

Potential biases in the training dataset

Generalizability across different patient demographics

Performance variations across different imaging equipment

Long-term validation requirements

The authors should consider adding technical comparisons with other interpretable architectures beyond standard DNNs. This would help readers understand the relative advantages and trade-offs of the sparse BagNet approach.

Despite these suggestions for improvement, the manuscript represents a significant contribution to the field. The combination of technical innovation, thorough validation, and demonstrated clinical benefit makes it suitable for publication in PLOS Digital Health. The recommended revisions would enhance what is already a valuable addition to the literature on AI in healthcare.

Regarding ethical considerations, the study appropriately obtained institutional review board approval (Ref No. 249/2023BO2) and handled patient data appropriately. The authors have made their implementation and study data available through GitHub, supporting reproducibility and further research.

In conclusion, I recommend acceptance with minor revisions. This work advances both the technical and clinical aspects of AI-assisted diabetic retinopathy screening, with particular strength in addressing the critical need for interpretability in clinical AI systems.

Reviewer #2: This manuscript presents a well-executed study on interpretable AI for diabetic retinopathy screening. The methodology is rigorous, with comprehensive validation and a strong reader study demonstrating clinical utility. While the statistical analysis and results are compelling, particularly regarding reduced screening time without accuracy loss, some minor revisions are needed. The data availability statement needs expansion with specific repository links, and the discussion of clinical implementation could be strengthened. The writing is clear with only minor grammatical issues to address. Overall, this is valuable work that, with these minor revisions, will be suitable for publication in PLOS Digital Health.

6. PLOS authors have the option to publish the peer review history of their article (what does this mean?). If published, this will include your full peer review and any attached files.

**Do you want your identity to be public for this peer review?** For information about this choice, including consent withdrawal, please see our Privacy Policy.

Reviewer #1: No

Reviewer #2: No

---

## [Decision Letter · Decision Letter 1]

19 Mar 2025

An Inherently Interpretable AI model improves Screening Speed and Accuracy for Early Diabetic Retinopathy

PDIG-D-25-00006R1

Dear Prof. Dr. Berens,

We are pleased to inform you that your manuscript 'An Inherently Interpretable AI model improves Screening Speed and Accuracy for Early Diabetic Retinopathy' has been provisionally accepted for publication in PLOS Digital Health.

Best regards,

Po-Chih Kuo, Ph. D.

Section Editor

PLOS Digital Health

**Additional Editor Comments (if provided):**

**Reviewer Comments (if any, and for reference):**

Reviewer's Responses to Questions

**Comments to the Author**

1. If the authors have adequately addressed your comments raised in a previous round of review and you feel that this manuscript is now acceptable for publication, you may indicate that here to bypass the “Comments to the Author” section, enter your conflict of interest statement in the “Confidential to Editor” section, and submit your "Accept" recommendation.

Reviewer #1: (No Response)

Reviewer #2: All comments have been addressed

2. Does this manuscript meet PLOS Digital Health’s publication criteria? Is the manuscript technically sound, and do the data support the conclusions? The manuscript must describe methodologically and ethically rigorous research with conclusions that are appropriately drawn based on the data presented.

Reviewer #1: Yes

Reviewer #2: Yes

3. Has the statistical analysis been performed appropriately and rigorously?

Reviewer #1: Yes

Reviewer #2: Yes

4. Have the authors made all data underlying the findings in their manuscript fully available (please refer to the Data Availability Statement at the start of the manuscript PDF file)?

Reviewer #1: Yes

Reviewer #2: Yes

5. Is the manuscript presented in an intelligible fashion and written in standard English?

Reviewer #1: Yes

Reviewer #2: Yes

6. Review Comments to the Author

Reviewer #1: The manuscript presents a significant advance in the use of interpretable artificial intelligence for screening diabetic retinopathy. The proposed approach, based on the Sparse BagNet model, demonstrates strong clinical potential by offering transparency in the decision-making process of artificial intelligence, differentiating itself from traditional deep learning methods.

Below, I present a detailed analysis of the main aspects of the study, considering positive points and suggestions for improvements.

1. Quality of Methodology and Validation

Comprehensive validation: The study was continuous with a robust database, composed of 34,350 images, and validated in 10 external datasets, ensuring good generalization of the model.

Retrospective clinical study: The participation of six ophthalmologists in the study reinforces the practical applicability of the model in medical decision-making.

Comparison with post-hoc techniques: The comparison with approaches such as Integrated Gradients and Guided Backpropagation confirms the superiority of the interpretability of the proposed model.

Suggestion: The study could benefit from a more detailed description of the choice of the sparsity hyperparameter (λ), better explaining the selections adopted to define this value.

2. Statistical Analysis

Accuracy and reliability: The study presents complete statistics, including 95% confidence intervals and the use of bootstrap with 1,000 resamplings, ensuring robustness in the analyses.

Proven clinical impact: The model increased diagnostic accuracy by 17.5% and reduced screening time by 25%, confirming practical benefits.

Suggestion: Although the statistical analysis is solid, it is recommended to include:

Retrospective statistical power calculation to specify the sample size of the retrospective clinical study.

Effect size calculation to quantify the magnitude of the observed improvements.

3. Data Availability and Reproducibility

Open source code: The model and analysis scripts have been made available on GitHub, ensuring transparency and reproducibility.

Public datasets: The images used provide open source databases, allowing replication of the study by other researchers.

Suggestion: To facilitate future validations, the authors were able to provide a structured guide on how to reproduce the experiments using the code made available.

4. Clinical Applicability and Integration

Low computational cost: The model is efficient and can be integrated into clinical systems without high hardware costs.

Impact on medical decision-making: The retrospective clinical study confirms that interpretable artificial intelligence improved the screening of diabetic retinopathy and reduced analysis time.

Suggestion: It is recommended to further detail:

Technical specifications for clinical implementation, including minimum hardware requirements and average inference time.

Guidelines for integration with existing clinical workflows and regulatory challenges for approval of the model in different countries.

5. Writing and Clarity of the Text

Clear and objective writing: The manuscript is well structured and technically sound.

Use of standardized terminology: Concepts are presented consistently and without ambiguity.

Suggestion: A final grammatical review can eliminate minor inconsistencies and improve the text's fluidity.

Conclusion and Final Recommendation

The study presents a relevant contribution to the field of artificial intelligence in health, combining high performance with interpretability. The results are well-founded and show a positive clinical impact.

Incorporating the suggested minor revisions will help to further strengthen the reproducibility and clinical applicability of the work. I recommend accepting minor revisions, as the proposed improvements are incremental and do not compromise the quality of the study.

Reviewer #2: The authors have adequately addressed all previous comments, and the manuscript meets **PLOS Digital Health's** publication criteria. The study is **methodologically rigorous**, with appropriate statistical analysis and validation across multiple datasets. The inclusion of a **reader study with ophthalmologists** strengthens its clinical relevance. The data and implementation details are **openly available**, ensuring reproducibility. The improvements in **model interpretability, screening accuracy, and efficiency** make this a valuable contribution to AI-assisted medical imaging. **No further concerns—recommended for acceptance.**

7. PLOS authors have the option to publish the peer review history of their article (what does this mean?). If published, this will include your full peer review and any attached files.

**Do you want your identity to be public for this peer review?** For information about this choice, including consent withdrawal, please see our Privacy Policy.

Reviewer #1: **Yes: **Francisco José Gonçalves Figueiredo

Reviewer #2: No
